# Automatic Speech Recognition for Uyghur, Kazakh, and Kyrgyz: An Overview

**Wenqiang Du** [1,†] , **Yikeremu Maimaitiyiming** [2,†], **Mewlude Nijat** [2], **Lantian Li** [3,*], **Askar Hamdulla** [2,*] and **Dong Wang** [1,*]

1   Center for Speech and Language Technologies, BNRist, Tsinghua University, Beijing 100084, China
2   School of Information Science and Engineering, Xinjiang University, Ürümqi 830017, China
3   School of Artificial Intelligence, Beijing University of Posts and Telecommunications, Beijing 100876, China
*   Correspondence: lilt@bupt.edu.cn (L.L.); askar@xju.edu.cn (A.H.);
    wangdong99@mails.tsinghua.edu.cn (D.W.); Tel.: +86-10–62796589 (D.W.)
†   These authors contributed equally to this work.

**Abstract:** With the emergence of deep learning, the performance of automatic speech recognition (ASR) systems has remarkably improved. Especially for resource-rich languages such as English and Chinese, commercial usage has been made feasible in a wide range of applications. However, most languages are low-resource languages, presenting three main difficulties for the development of ASR systems: (1) the scarcity of the data; (2) the uncertainty in the writing and pronunciation; (3) the individuality of each language. Uyghur, Kazakh, and Kyrgyz as examples are all low-resource languages, involving clear geographical variation in their pronunciation, and each language possesses its own unique acoustic properties and phonological rules. On the other hand, they all belong to the Altaic language family of the Altaic branch, so they share many commonalities. This paper presents an overview of speech recognition techniques developed for Uyghur, Kazakh, and Kyrgyz, with the purposes of (1) highlighting the techniques that are specifically effective for each language and generally effective for all of them and (2) discovering the important factors in promoting the speech recognition research of low-resource languages, by a comparative study of the development path of these three neighboring languages.

**Keywords:** overview; automatic speech recognition; low-resource; Uyghur; Kazakh; Kyrgyz

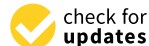



## 1. Introduction

According to the statistics of the Ethnologue website (https://www.ethnologue.com/, accessed on 25 December 2022), there are 7151 known living languages in the world (every living language has at least one speaker for whom it is his/her first language). Most of these are low-resource languages, which refers to a language with some (if not all) of the following aspects [1]: a lack of a unique writing system or a stable orthography, a limited presence on the web, a lack of linguistic expertise, and/or a lack of electronic resources for speech and language processing, such as monolingual corpora, bilingual dictionaries, transcribed speech data, pronunciation dictionaries, vocabularies, etc.

Information-processing technology for low-resource languages is of great significance to strengthen culture exchanges, promote information fairness, and protect endangered languages. Automatic speech recognition (ASR) is among the important technologies in language information processing. After several decades of development, modern ASR systems, especially for rich-resource languages, have achieved significant performance improvements and have been deployed in a wide range of practical applications. However, the performances for low-resource languages are still far from satisfactory.

In general, ASR for low-resource languages encounters three challenges: (1) the data scarcity of speech and text, leading to difficult model training; (2) a lack of standardization, a discrepancy in the pronunciation, and linguistic changes, leading to the uncertainty of the

languages. (3) each language having its own individual properties, reflected in the writing, linguistic, phonetic, acoustic, and other aspects. If these challenges cannot be solved well, it would be impossible to achieve reasonable recognition performance. Currently, the vast majority of research focuses on the data scarcity, while less attention has been paid to the standardization and uniqueness of the language itself.

Uyghur, Kazakh, and Kyrgyz are all low-resource languages and possess the challenges mentioned above. However, they all belong to the Altaic branch of the Altaic language family, and so, they have a large degree of similarity at both the acoustic and linguistic levels. Especially in China, the three languages are distributed across neighboring regions, which closely interact with each other, so they are quite similar in their writing, pronunciation, and syntax [2]. From the perspective of ASR, this means that similar techniques, especially those that can be used to tackle the challenges of low-resource languages, can be used for all three.

Therefore, this paper investigates these three languages together and summarizes the modern algorithms/schemes for each language. Our first goal was to highlight the important techniques that have been broadly verified so that researchers working on different languages can learn from each other. In addition, by comparing the technical roadmap of these three neighboring languages, we hope to discover the key factors for developing low-research language speech recognition research, thereby providing some suggestions on the "important work" that researchers may focus on.

The rest of this paper is organized as follows: Section 2 presents an overview of low-resource language speech recognition methods. Section 3 provides a comprehensive linguistic analysis of Uyghur, Kazakh, and Kyrgyz and summarizes their commonalities and individualities. A technical review, analysis, and discussion of the speech recognition techniques for these three languages are reported in Section 4. Section 5 concludes the paper.

## 2. Low-Resource Language Speech Recognition

### 2.1. History of Speech Recognition Technology

Speech recognition technology has been developing for half a century. Traditional speech recognition systems are based on a Gaussian mixture model-hidden Markov model (GMM-HMM) methodology [3]. This involves a GMM to represent the distribution of a stationary state and an HMM to represent a state transition. This method is essentially a generative model, which describes the generation process of the speech signals and resorts to Bayesian inference to determine the spoken words.

In recent years, deep learning methods have demonstrated significant success in speech recognition. There are two commonly used methods: the deep neural network-hidden Markov model (DNN-HMM) and the end-to-end (E2E) model. The former retains the framework of the generative model, while the latter is purely discriminative.

For the DNN-HMM method, a deep neural network (DNN) replaces the GMM to determine the likelihood of each frame for each HMM state [4,5]. Compared to the GMM-HMM, a key advantage of the DNN-HMM is that the DNN can extract the abstract features from the raw data, leading to better discrimination and generalizability. The initial success of the DNN motivated more effective neural models, including convolutional neural networks (CNNs) [6–10] and recurrent neural networks (RNNs) [11–15]. Compared to the standard fully connected DNN, the CNN and RNN are more effective at utilizing the temporal invariance of the speech signals.

For the E2E method, it regards speech recognition as a sequence-to-sequence generation task. Specifically, it uses language information in the acoustic modeling and achieves the joint learning of the acoustic model and the language model. Since the E2E structure unifies all the components of speech recognition into one neural network model, it omits the intermediate manual design processes (e.g., acoustic features, pronunciation units, etc.). Especially, when the training data are rich enough, a better recognition performance can be obtained. Two typical E2E structures are based on connectionist temporal classification (CTC) loss functions [12,16] and sequence-to-sequence modeling [17,18].

The central idea of the CTC loss function is to maximize the likelihood that the input speech sequence produces the output text sequence, by marginalizing all possible alignments. To further strengthen the explicit learning of language models, the recurrent neural network transducer (RNN-T) method is generally adopted to learn the context dependencies among output labels to achieve better performance [19]. The sequence-to-sequence model directly simulates the behavior of human listening and speaking and describes the recognition process as a language generation process with the input speech as a reference. Most of these models are based on a mechanism called *attention*, which focuses on the speech segments related to the current decoding state in the process of recognition.

In recent years, researchers have proposed the self-attention model. On the one hand, this model is based on the attention mechanism, so it can automatically discover significant information in the sequence during the feature extraction process. On the other hand, it supports long-term sequence modeling in parallel, thus eliminating a major problem for traditional temporal models such as the RNN. A representative self-attention model is *Transformer* [20], which is widely used in natural language processing (NLP) tasks [21–24] and has been successfully migrated to the speech recognition field [25–27].

In the past two years, speech pre-training models based on self-supervised learning (SSL) have received considerable attention. By designing a reasonable proxy task, this approach can utilize a large amount of unannotated speech data to train a base model, i.e., the pre-training model [28–30]. Considering that there is a strong correlation among most speech-processing tasks—for instance, speech denoising is necessary for both speech recognition and speaker recognition and extracting the typical patterns is important for both keyword spotting and speech coding—the pre-trained model can be applied to various downstream speech-processing tasks. Wav2Vec [28,29] and HuBERT [30] are the two most-popular speech pre-training models.

Generally speaking, due to the accumulation of large amounts of training data and the powerful capacity of deep neural networks, speech recognition for common languages, e.g., English and Chinese, has achieved practical performance in many applications. For the traditional and DNN-based speech recognition technologies, please refer to [31,32] and [3,33–35], respectively, and the latest advances can be found in [5,36,37].

### 2.2. Methods of Low-Resource Language Speech Recognition

As mentioned above, low-resource language speech recognition encounters three issues: data scarcity, uncertainty in writing and pronunciation, and the individuality of each language. Most of the current research focuses on the data scarcity. The intuitive way to solve this issue is to collect more data. Recently, crowdsourcing has become a popular way to collect a large volume of data at a low cost [1,38]. Despite the clear importance of data collection, in this review paper, we focus on technical approaches to the low-resource language problem and summarize several representative methods, as shown in Table 1.

**Table 1.** Representative methods of low-resource ASR.

| Method | Summary |
| --- | --- |
| Data augmentation | Various data augmentation approaches to simulate complex behavioral and/or environmental variety. |
| Phoneme mapping | Construct correlations between phonemes of different languages, so that one language can borrow models from other languages. This may construct a system for a new language even without any training data. |
| Feature sharing | Resorting to the commonality of human pronunciation, employ multilingual data to train generic acoustic feature extractor, so that the data required for training the acoustic model for a new language can be significantly reduced. |

**Table 1.** *Cont.*

| Method | Summary |
|---|---|
| Unsupervised learning | Using semi-supervised or self-supervised learning approach to utilize unlabelled data. |
| Completely unsupervised learning | No transcribed data at all, employ unparallel speech and text to train ASR systems. |
| Massively multilingual modeling | Utilize speech data and lexical knowledge of a large amount of languages, to train models for 'any' language (nearly). |

### 2.2.1. Data Augmentation

Data augmentation generates diverse data by designing various transformation functions [39,40]. For speech recognition, popular data augmentation methods include adding noise [39] and reverberation [41], speed perturbation [42], using vocal tract length perturbation to generate warped spectral variants [43], speech synthesis [44–47], SpecAugment [48], MixSpeech [49], sub-sequence sampling [50], etc. Data augmentation is one of the most-effective methods for low-resource language speech recognition and has been widely used in practice [40,51–54].

### 2.2.2. Phoneme Mapping

There are many similar pronunciation units (e.g., phonemes) among different languages. If the phonemes of different languages can be unified into a universal phoneme set, it would be easy to set up a phoneme map between two languages. Based on this mapping, models of other languages can be used to "synthesize" the model of the target language. This method is especially suitable for acoustic models based on the GMM, as GMMs corresponding to different phonemes are independent. For example, Schultz et al. [55] used the International Phonetic Alphabet (IPA) to accomplish the mapping between languages. With this mapping, they used the acoustic models of seven languages to construct a Swedish speech recognition system and obtained good results. Lööf et al. [56] adopted the same method to transfer a Spanish model to a Polish model.

### 2.2.3. Feature Sharing

With the emergence of deep learning, feature sharing becomes a standard and primary approach for low-resource language speech recognition. The basic idea of this method is that human languages are similar in speech production so that the acoustic features of different languages can be largely shared. Three approaches are commonly used to share the features in practice [57]: (1) to learn a language-independent feature extraction network and, then, use the network to extract features for low-resource languages. This approach is mostly applied in the tandem ASR framework [58], and it firstly trains a DNN model with rich-resource language monolingual or multilingual data, then it uses this DNN model to extract language-independent acoustic features, and finally, it trains the GMM-HMM or DNN-HMM for low-resource languages [59–63]. (2) Multilingual learning: This approach is mostly based on the hybrid ASR framework [31]. The main architecture is a multi-head DNN trained with the speech data of multiple languages including the target language, where the feature extractor is shared and each language maintains its own phoneme recognition layer [57,64]. (3) Transfer learning: This firstly uses rich-resource language monolingual or multilingual data to train a neural network, and then, it uses this to initialize the model for the low-resource language. A fine-tuning step then follows to adapt the model to the data of the low-resource language [65].

In recent years, end-to-end speech recognition technology has grown popular, and how to transfer end-to-end models to low-resource language speech recognition has become a research focus. Dalmia et al. [66] trained a CTC model with the multilingual data in the BABEL dataset, showing that the multilingual training improved the performance

for each language. Inaguma et al. [67] built a multilingual attention model. They trained this attention model using the data from 10 languages in the BABEL dataset and, then, migrated it to the other five languages in BABEL. Similar to the scheme proposed by [67], Watanabe et al. [68] constructed a multilingual CTC-attention model. Cho et al. [69] adopted this multilingual CTC-attention model and reported better results than monolingual models when experimented on the BABEL dataset. Zhou et al. [70] verified the performance of multilingual Transformer models. It was based on multilingual character sets and used the BPE algorithm to generate subword units (https://github.com/rsennrich/subword-nmt, accessed on 25 December 2022). Experimental results on six languages in the CALLHOME dataset showed that the Transformer model trained on multilingual data could achieve better results over monolingual models. Shetty et al. [71] also verified the multilingual Transformer model, and in particular, they found that involving language information can further improve the model's strength. Müller et al. [72] also demonstrated the importance of language information. They added a learnable language vector to the DNN and found that the performance on low-resource languages was improved.

The basic assumption of feature sharing is that the different languages are similar in speech generation. This idea is not new and was also studied in the context of the traditional GMM-HMM framework. For example, Burget et al. [73] divided the parameters of the subspace Gaussian mixture models (SGMMs) [74] into two parts: language-independent global parameters and language-dependent local parameters, and used the multilingual data to train the global parameters and the monolingual data to train the local parameters, obtaining significant performance improvement through this multilingual training.

It is worth mentioning that some studies proposed using a universal phoneme set to train multilingual DNN models, which is essentially a combination of phoneme sharing and feature sharing. For example, Vu et al. [75] and Tong et al. [76] built a unified annotation for different languages based on the IPA and, then, trained the DNN-HMM, multilingual CTC, and lattice-free maximum mutual information (LF-MMI) models [77]. Kim et al. [78] combined the characters of English, German, and Spanish as a universal character set and trained the CTC model based on this set. An obvious advantage of this phoneme-sharing model is that it can be transferred to a zero-resource model. In principle, if the phonemes of a new language are covered by those of the languages in the training set, the transfer of the zero-resource model is possible. Even if some phonemes are not covered, similar phonemes from the training languages can be used as a substitution [79].

2.2.4. Unsupervised Learning

Unsupervised learning is another important method to solve the data scarcity. The basic idea of this method is to learn the shared representations through a large number of unlabeled data, thus reducing the data requirement of acoustic modeling.

Semi-supervised training is a typical unsupervised learning approach. In this training scheme, an initial model is firstly trained with a small amount of annotated data, and the model is used to decode the unlabeled data. Once completed, the resultant transcripts are treated as pseudo-labels and used to retrain the model [80]. Billa et al. [81] recently found that semi-supervised learning can improve the performance of low-resource language speech recognition even with data from mismatched domains. More aggressively, semi-supervised training can be conducted in a cross-lingual manner, which decodes unlabeled data by an off-the-shelf model of a rich-resource language and, then, translates the resultant transcripts to the target language, e.g., by phoneme mapping [82,83].

Self-supervised training is perhaps the most-popular unsupervised learning approach currently. For example, Conneau et al. [84] trained a Wav2Vec 2.0 model with 800 h of speech data in 10 languages. After fine-tuning on the BABEL data, this model achieved remarkable performance improvement in multilingual tests. Javed et al. [85] reproduced this result on a database containing 40 Indian languages from a wide variety of domains including education, news, technology, and finance. Zhang et al. [86] also adopted the

multilingual self-supervised training scheme in low-resource language settings, though their model was based on a teacher–student learning scheme.

Besides pre-training, unsupervised learning also can be used as regularization. For example, Renduchintala et al. [83] proposed a multi-modal data augmentation (MMDA) method, which feeds text to the encoder of a sequence-to-sequence model and, then, lets the decoder reconstruct the input text. In this setting, the model is trained with extra text data. This extra training is unsupervised and plays the role of regularization. Wiesner [87] improved this approach by passing the text through a pre-processing network before feeding it to the encoder. Wang et al. [88] proposed a UniSpeech model, which treated the model training of Wav2Vec 2.0 as an auxiliary task to obtain robust speech representations at the encoding layers.

### 2.2.5. Completely Unsupervised Learning

Recently, researchers have been paying attention to speech recognition methods under the conditions of extreme data scarcity; for example, there are unlabeled speech and independent text, but no transcribed speech is available. In this case, the training is completely unsupervised. Liu et al. [89] firstly proposed a completely unsupervised learning method that combines model pre-training with adversarial learning. This method extracts speech representations based on Wave2Vec 2.0 and, then, generates phoneme sequences based on a generative adversarial network (GAN) model, which includes a generator and a discriminator, iteratively learning from each other. The training objective is to make the distributions of the phoneme sequences generated by the GAN match the distribution of the real text. This seminal work was followed by a couple of researchers [90–92]. Completely unsupervised learning is a hotspot at present, and its latest progress can be found in [93].

### 2.2.6. Massively Multilingual Modeling

Thanks to the accumulation of data and the development of techniques, researchers have been able to carry out massively multilingual modeling. In 2019, researchers at Johns Hopkins University studied the possibility of pre-training speech models using the data from a huge amount of languages [94]. They trained multilingual models using the Bible speech data of 100 languages in the CMU multilingual database and used the multilingual models to initialize the models of other languages [95]. Later, researchers at Facebook trained a multilingual model using data from 50 languages [96]. Although fewer languages were used, their data were beyond the Bible data and much more complex. Recently, researchers at CMU [97] trained speech recognition systems for 1909 languages. They trained a large-scale acoustic model on English and then mapped the phonemes of other languages to the phonemes of the English model based on linguistic knowledge (such as PHOIBLE [98]).

### 2.2.7. Other Resources

Current databases for low-resource language speech recognition include Globalphone [99], BABEL [100], the CMU Wild Multilingual Database [95], VoxForge (http://www.voxforge.org/, accessed on 22 December 2022), Multilingual LibriSpeech (MLS) [101], Common Voice [102], etc. For low-resource language speech recognition, early approaches can be found in [1] and recent advances can be found in [103,104].

## 3. Linguistic Analysis of Uyghur, Kazakh, and Kyrgyz

These three languages are so related that there are usually no barriers to communication for speakers among each other [105]. Even so, each language also has its individualities.

### 3.1. Commonalities of the Three Languages

From the writing aspect, the three languages are all written from right to left and the words are separated by blanks. In history, the three languages all underwent abrupt changes in their writing systems. For example, Kyrgyz was written in ancient Altaic letters and then

gradually changed to Persian-Arabic letters. After the establishment of the Soviet Union (The USSR), Kyrgyz in The USSR began to use Latin letters from 1927 and Cyrillic letters from 1940. After the collapse of The USSR, people in Kyrgyzstan discussed the possibility of changing back to Latin letters, but to today, Cyrillic letters still dominate in Kyrgyzstan. In China, Kyrgyz is relatively stable and is always written in Arabic letters [106]. The other two languages have also experienced similar changes. In China, all three languages use Arabic letters, though a slight difference exists. The following discussion will focus on the three languages in China.

From the linguistic aspect, the syntax rules of the three languages are generally the same and a large proportion of words look similar. In morphology, they share the same feature of suffix agglutination, a prominent characteristic of Altaic languages. Specifically, each word is composed of a stem as a skeleton plus multiple suffixes (additional components), and some words also contain a prefix. The agglutination is rather flexible, and the number of suffices could be as large as 10, making the vocabulary very large. Taking Uyghur as an example, there are about 40,000 common stems and 289 affixes, constructing more than 1.2 million words [107]. Despite the large vocabulary, the letter-to-phone conversion is rather simple, and in most cases, it is a one-to-one mapping.

From the phonetic aspect, each word consists of several syllables and each syllable should contain a vowel. All three languages follow strict rules of phonetic harmony when syllables are combined to form words. There are two types of phonetic harmony: vowel harmony and consonant harmony. Vowel harmony means that the vowels appearing in different syllables of the same word are consistent in the articulation position and lip shape, that is either all are front vowels (or back vowels) or all are rounded vowels (or unrounded vowels). Consonant harmony means that consonants in successive syllables are either all voiced or all voiceless [108].

From the acoustic aspect, the three languages are also similar. Experiments have shown that the patterns of the fundamental frequency (F0) of the three languages are quite similar for females and males. Besides, the averaged values of the first formant (F1) of the corresponding vowels in the three languages are substantially the same, while the difference is mainly in the second formant (F2): the F2 of Kazakh and Kyrgyz is clearly higher than that of Uyghur [109].

### 3.2. Individualities of Each Language

#### 3.2.1. Alphabets

Kazakh and Kyrgyz are based on Arabic letters, while Uyghur is slightly different. Specifically, 20 letters are identical, 2 letters are spelled in different ways, and 2 letters are unique in Uyghur (see Appendix D).

#### 3.2.2. Phoneme Set

All three languages were developed from the ancient Altaic, and their phoneme sets are similar. Kyrgyz best preserves the ancient Altaic phonemes, while Uyghur and Kazakh have developed new phonemes. Table 2 summarizes the vowels used in the three languages, and the consonants are listed in the Appendix section.

**Table 2.** Vowel phoneme sets of Uyghur, Kazakh, and Kyrgyz.

| Vowels | e | a | ø | o | u | y | æ | ε | ɨ | i | ə |
|---|---|---|---|---|---|---|---|---|---|---|---|
| Front (←)/Central (•)/Back (→) | ← | → | ← | → | → | ← | ← | ← | • | ← | • |
| Top (↑)/Bottom (↓) | ↓ | ↓ | ↓ | ↓ | ↑ | ↑ | ↓ | ↑ | ↑ | ↑ | ↑ |
| Rounded (o)/Unrounded (¬o) | ¬o | ¬o | o | o | o | o | ¬o | ¬o | ¬o | ¬o | ¬o |
| Uyghur | ✓ | ✓ | ✓ | ✓ | ✓ | ✓ | | ✓ | | ✓ | |
| Kazakh | ✓ | ✓ | ✓ | ✓ | ✓ | ✓ | ✓ | | ✓ | | ✓ |
| Kyrgyz | ✓ | ✓ | ✓ | ✓ | ✓ | ✓ | | | | ✓ | ✓ |

Specifically, in Kyrgyz, there are 8 short vowels: *a, e, ə, o, ø, u, i,* and *y,* and 22 consonants (see Appendix C). The eight short vowels are rooted in the ancient Altaic, and based on these basic vowels, Kyrgyz extended six long vowels: *aa, oo, uu, ee, øø,* and *yy.* These long vowels may be formed by the consonant assimilation between vowels [110].

Kazakh has nine vowels and 24 consonants (see Appendix B). Apart from the same vowels as Kyrgyz, Kazakh has an extra front vowel *æ,* and the front vowel *y* is replaced by a central vowel *ɨ.* Some researchers believe that *æ* in Kazakh did not appear in the ancient Altaic, instead gradually emerging in the process of language fusion [111]. Besides the vowels, Kazakh also has 24 consonants (see Appendix B).

Uyghur is based on eight vowels: *a, ɛ, e, i, o, ø, u,* and *y,* and 24 consonants [110] (see Appendix A). The Uyghur vowels are very similar to the Kyrgyz vowels, except that the central vowel *ə* was substituted with the front vowel *ɛ.*

### 3.2.3. Vowel Harmony

As mentioned above, vowel harmony exists in all three languages, but the degree of strictness is different. For Kyrgyz, the vowel harmony rule is the most strict. For Kazakh, the harmony rule may be broken into words involving the vowel *æ,* perhaps because it is a new pronunciation in the language. For Uyghur, *e* and *i* are relatively neutral, and so can be combined with either front vowels or back vowels [110].

### 3.2.4. Vowel Reduction

Vowel reduction refers to the "weakening" effect of vowels when suffixes are added. With the weakening effect, an open vowel will be changed to a closed vowel (position from bottom to top). For example, in Uyghur, *a* weakens to *e*: *ati ← eti* ("his horse") and *ɛ* weakens to *e*: *bɛli ← beli* ("her waist"). The vowel reduction of *a* and *ɛ* is a unique characteristic of Uyghur.

### 3.2.5. Other Differences

Besides the individual properties of the three languages mentioned above, other differences include: (1) the position of each vowel and consonant in word formation; (2) the harmony rule of consonants and the weakening rules (inspiration of aspirated phonemes, voiceless of voiced consonants, spirantization of stops and affricate consonants, spirantization of semivowels, e.g., *y*[*j*], etc.); (3) the elision of vowels and consonants; (4) the impact of foreign words on acoustic and linguistic rules.

## 4. Advances in Speech Recognition for Uyghur, Kazakh, and Kyrgyz

### *4.1. Resource Accumulation*

Resources comprise the major bottleneck that restricts the research of low-resource language speech recognition. To solve the resource problem, many previous research institutions designed various databases and language resources (dictionaries, texts, language models, etc.). Most of these resources are for internal usage and have poor standardization. Results based on private data are not convincing enough. Recently, researchers noticed this problem and have published several open databases, which has greatly promoted the progress of the research. Table 3 summarizes the datasets available at present.

### 4.1.1. THUYG-20

THUYG-20 [112] is a Uyghur speech recognition database published by Tsinghua University and Xinjiang University in 2017. It consists of 20 h of training speech recorded by 348 speakers (163 males and 185 females) and 2 h of test speech recorded by 23 speakers (13 males and 10 females). The recording device is a table microphone connected to a desktop. All the speech data were collected in an office environment, and the speakers uttered the sentences in the reading style. Besides speech data, the research group also published dictionaries (phoneme dictionary and morpheme dictionary), language models, and all the source codes (https://github.com/wangdong99/kaldi/tree/master/egs/

thuyg20, accessed on 25 December 2022). THUYG-20 is the first complete and open-source Uyghur database.

### 4.1.2. M2ASR

In 2016, Tsinghua University, Xinjiang University, and Northwest Minzu University kicked off a joint project called Multilingual Minorlingual Automatic Speech Recognition (M2ASR). The goal of this project was to conduct research on speech recognition technologies for five minority languages, including Tibetan, Mongolia, Uyghur, Kazakh, and Kyrgyz [113]. At the end of this project in 2021, the project team released all the resources they constructed. The resources included speech databases (277 h in Uyghur, 250 h in Kazakh, and 166 h in Kyrgyz), text corpora (88 M in Uyghur, 100 M in Kazakh, and 6 M in Kyrgyz), and pronunciation dictionaries (45,000 in Uyghur, 100,000 in Kazakh, and 15,000 in Kyrgyz). This is the most-comprehensive and -complete resource for Uyghur, Kazakh, and Kyrgyz speech recognition.

### 4.1.3. Common Voice

Common Voice is a free and open crowdsourced data collection project initiated by Mozilla and other companies and research institutes [102]. Volunteers upload speech to the cloud server, and then, anyone can perform annotation via the same platform. In 2019, Common Voice 1.0 was released. It contained 1000 h of speech from 19 languages. At present, Common Voice has been updated to Version 11.0 (21 September 2022), covering 100 languages and containing 16,000 h of speech. In the new release, there are 119 h of Uyghur speech, 2 h of Kazakh speech, and 47 h of Kyrgyz speech. Common Voice is the first multilingual data resource constructed by crowdsourcing, and it is continually accumulating data. For now, the Kyrgyz data in Common Voice is the standard data for Kyrgyz research.

### 4.1.4. KSC/KSC2

In 2020, researchers at Nazarbayev University in Kazakhstan published a 330 h Kazakh speech corpus (KSC) [114], which was the largest Kazakh database at that time. Motivated by Common Voice, this database was also collected by crowdsourcing. The entire dataset contains more than 150,000 speech utterances from 1612 devices. In the same year, this research group published a Kazakh TTS database, called KazakhTTS [115]. It was recorded by two speakers (one male and one female) and comprises 93 h in total. In 2022, this group expanded KazakhTTS to five speakers and 271 h of speech, called KazakhTTS2 [116]. Recently, KSC and KazakhTTS2 were put into a larger dataset called KSC2 [117], together with additional data including: (1) 238 h of crowdsourcing data; (2) 48 h of high-quality TTS speech; (3) 238 h of speech from TV, self-media, and other sources. Now, KSC2 has reached 1128 h in total. It is the largest Kazakh database at present.

**Table 3.** Active resources for Uyghur, Kazakh, and Kyrgyz.

| Resource | Language | Date | Contributor | Data Size | Attributes |
|---|---|---|---|---|---|
| THUYG-20 [112] [1] | Uyghur | 2017 | THU & XJU | Speech: 20 h | Reading, Office, Mic |
| M2ASR-Uyghur [2] | Uyghur | 2021 | M2ASR | Speech: 136 h, Text: 88 M | Reading, Mobile |
| M2ASR-Kazakh [118] [2] | Kazakh | 2021 | M2ASR | Speech: 78 h, Text: 100 M | Reading, Mobile |
| M2ASR-Kyrgyz [2] | Kyrgyz | 2021 | M2ASR | Speech :166 h, Text: 6 M | Reading, Mobile |
| CommonV-Uyghur [102] [3] | Uyghur | 2021 | Mozilla, etc. | Speech: 119 h | Crowdsourced |
| CommonV-Kazakh [102] [3] | Kazakh | 2021 | Mozilla, etc. | Speech: 2 h | Crowdsourced |
| CommonV-Kyrgyz [102] [3] | Kyrgyz | 2021 | Mozilla, etc. | Speech: 2 h | Crowdsourced |
| IARPA-Kazakh [100] [4] | Kazakh | 2021 | IARPA | Speech: 64 h | Mic and Telephone |

**Table 3.** *Cont.*

| Resource | Language | Date | Contributor | Data Size | Attributes |
|---|---|---|---|---|---|
| KSC [114] [5] | Kazakh | 2020 | ISSAI | Speech: 332 h | Crowdsourced |
| KazakhTTS [115] [6] | Kazakh | 2021 | ISSAI | Speech: 93 h | 2 Speakers |
| KazakhTTS2 [116] [6] | Kazakh | 2022 | ISSAI | Speech: 271 h | 5 Speakers |
| KSC2 [117] [7] | Kazakh | 2022 | ISSAI | Speech: 1128 h | Crowdsourced, Reading |

[1] http://openslr.org/22 (Free). [2] http://m2asr.cslt.org (Free). [3] https://commonvoice.mozilla.org/en/datasets (Free). [4] https://catalog.ldc.upenn.edu/LDC2018S13 (Not free). [5] https://issai.nu.edu.kz/kz-speech-corpus (Free). [6] https://github.com/IS2AI/Kazakh_TTS (Free). [7] https://github.com/IS2AI/Kazakh_ASR (Free). All the resources were accessed on 25 December 2022.

### 4.2. Technical Development

#### 4.2.1. Uyghur

Speech recognition in Uyghur started quite early. Wang et al. constructed a Uyghur database in 1996 [119] and, then, performed an in-depth study of Uyghur speech recognition techniques. For instance, in 2003, they studied the performance of different recognition units under the HMM framework and found that syllables were suitable units for acoustic modeling [120]. Silamu et al. [121] discussed the construction principle of the Uyghur database in 2009 and also studied the speech recognition approach based on the HMM and the speech synthesis approach based on unit selection. In 2010, Ablimit et al. [122] compared two Uyghur speech recognition systems based on morphemes and words, respectively. Experiments conducted on 150 h of microphone data from 353 speakers showed that the performance of the word-based system was superior. In the same year, Li et al. [123] built a large-scale Uyghur speech recognition system based on stem–suffix. Experiments conducted on 500 h of telephone data from 835 speakers found that the system based on stem–suffix obtained better performance than the system based on words.

After 2015, deep neural networks began to be applied to Uyghur speech recognition. For example, Tuerxun et al.[124] trained a DNN-HMM model with 50 h of data collected by USTC and reported better performance than the GMM-HMM (21.82%–>12.98%). Batexi et al. [125] trained a DNN-HMM model with 4466 utterances collected privately and reported more significant performance gains over the GMM-HMM (40.18%–>9.09%). In 2020, Ding et al. [126] built a novel end-to-end speech recognition system based on the Transformer+CTC architecture with the King-ASR450 database. In 2021, Subi et al. [127] implemented a Conformer system. Since then, the ASR research on Uyghur has kept up with the technical advance of common languages.

In addition to following state-of-the-art technology, researchers have also made some special designs according to the individualities of Uyghur.

Firstly, to solve the data scarcity issue, researchers put forward several schemes. In 2017, Yolwas et al. [128] verified the value of transfer learning in Uyghur speech recognition. They trained a basic BLSTM-HMM model with Chinese data and, then, retrained the output layer with Uyghur data. Since the volume of Uyghur data was already large, the contribution of the transfer learning was not clear. However, significant improvement was soon observed in data sparsity scenarios [129,130].

Another representative work towards data efficiency was conducted by Shi et al. [131] based on semi-supervised training. They firstly employed the transfer learning method to convert a Chinese model into a Uyghur model, by fine-tuning with a small amount of annotated Uyghur data. The obtained model was then used to decode a large amount of unlabeled Uyghur data, and the decoding results were used to refine the Uyghur model. They found that, with this method, using only 500 annotated utterances, they could achieve more competitive performance than the model trained from scratch with 50 h of annotated data.

Secondly, since the vocabulary of Uyghur is very large, researchers have proposed to use subword units in language modeling (LM). Ablimit et al. [132] developed a multi-

lingual morpheme analysis tool, which can discover frequent morphemes from a list of words. This tool supports Uyghur, Kazakh, and Kyrgyz. Rouzi et al. adopted this tool to build a morpheme-based Uyghur system and found that using morpheme-based LM can achieve better performance than word-based LMs [112]. In 2018, Hu et al. [133] proposed a subword-word end-to-end ASR model and reported good performance on a private Uyghur database involving 1000 h of speech. Further research showed that combining subword units and BPE encoding usually offered good performance in Uyghur ASR [134,135].

It is worth noting that most of the above studies were based on private data. An obvious consequence is that the claimed experimental results cannot be reproduced, making the research unverifiable. In addition to the inaccessible private database, the lack of open-source code is another obstacle for research development. This makes it difficult for readers to reproduce the blend of techniques, thus being unable to verify their effectiveness and innovation. To solve these two problems, a standard benchmark database and its accompanying baselines are required. In 2017, Tuersun et al. [136] described how they constructed a database of 1000 h of Uyghur, but neither the source code, nor the collected data were published. In the same year, Tsinghua University and Xinjiang University published a free Uyghur database, THUYG-20 [112], and also released several baselines with popular techniques, making technology comparison and discussion possible. Since then, many researchers have reported substantial progress based on this open database [127,129,135,137–139]. Table 4 shows the performance of different technologies reported on THUYG-20, from which we can clearly see the roadmap of technical progress. In this table, the DNN-HMM is the hybrid DNN system; Chain-TDNN is a variant of the hybrid system with a 1D temporal dilated CNN employed; BLSTM-CTC/attention is an end-to-end system, with both the CTC loss and the attention-based sequence-to-sequence architecture; Transformer-CTC and Conformer-CTC are both based on CTC training with Transformer and Conformer as the encoder blocks, respectively; and Conformer-CTC-MTL introduces multi-task learning on the basis of Conformer-CTC.

**Table 4.** SOTA results tested on THUYG-20 [127].

| Models | CER (%) |
| --- | --- |
| DNN-HMM | 24.3 |
| Chain-TDNN | 17.6 |
| BLSTM-CTC/Attention | 31.5 |
| Transformer-CTC | 21.4 |
| Conformer-CTC | 11.6 |
| Conformer-CTC-MTL | 7.8 |

### 4.2.2. Kazakh

Compared to Uyghur, research on Kazakh speech recognition started much later. In 2015, Khomitsevich et al. [140] studied the Kazakh–Russian bilingual speech recognition system. They built a Kazakh–Russian bilingual keyword recognition system based on the DNN-HMM structure with 147 h of telephone data. In 2016, Abilhayer et al. [141] constructed a continuous speech recognition system based on the GMM-HMM. Since 2019, the Institute of Information and Computational Technology and al-Farabi Kazakh National University published several papers on Kazakh speech recognition and verified the DNN-HMM system [142,143], BLSTM-CTC end-to-end system [144,145], and Transformer CTC/attention system [146] with their private data. At the same time, Beibut et al. [147] constructed an LSTM-CTC end-to-end Kazakh ASR system based on transfer learning. Since these research works were based on private data, their value is limited.

Like Uyghur, researchers have gradually become aware of the importance of open databases for Kazakh ASR. In 2021, Kuanyshbay et al. [148] reported their crowdsourcing platform and claimed that 50 h of data had been collected through this platform. However, they did not publish the platform, nor the data. In 2021 also, researchers from Nazarbayev

University released the first large-scale Kazakh database, KSC [114], offering the first open benchmark for Kazakh speech recognition research. Since then, research on Kazakh has been in the fast lane. For example, Mussakhojayeva et al. [149] conducted a multilingual study of Kazakh, Russian, and English based on KSC. In 2022, KSC was further expanded and reached 1128 h [117]. Table 5 presents the performance of different models tested on KSC, from which we can see the roadmap of technical development. In this paper, the DNN-HMM is the hybrid system, and the others are two end-to-end systems with LSTM and Transformer as the encoder blocks, respectively. Valid and Test represent the validation set and test set, respectively.

**Table 5.** SOTA results tested on KSC [114]. LM represents the language model; SpeedPerturb and SpecAug represent speed perturbation and SpecAugment, respectively.

| ID | Models | LM | SpeedPerturb | SpecAug | Valid | | Test | |
|---|---|---|---|---|---|---|---|---|
| | | | | | CER (%) | WER (%) | CER (%) | WER (%) |
| 1 | DNN-HMM | Yes | Yes | No | 5.2 | 14.2 | 4.6 | 13.7 |
| 2 | | Yes | Yes | Yes | 5.3 | 14.9 | 4.7 | 13.8 |
| 3 | E2E-LSTM | No | No | No | 9.9 | 32.0 | 8.7 | 28.8 |
| 4 | | Yes | No | No | 7.9 | 20.1 | 7.2 | 18.5 |
| 5 | | Yes | Yes | No | 5.7 | 15.9 | 5.0 | 14.4 |
| 6 | | Yes | Yes | Yes | 4.6 | 13.1 | 4.0 | 11.7 |
| 7 | E2E-Transformer | No | No | No | 6.1 | 22.2 | 4.9 | 18.8 |
| 8 | | Yes | No | No | 4.5 | 13.9 | 3.7 | 11.9 |
| 9 | | Yes | Yes | No | 3.9 | 12.3 | 3.2 | 10.5 |
| 10 | | Yes | Yes | Yes | 3.2 | 10.0 | 2.8 | 8.7 |

### 4.2.3. Kyrgyz

In contrast, research on Kyrgyz speech recognition started even later. Until 2018, researchers [150] from Xinjiang University reported their study based on a CNN model. In this study, they trained an initial CNN model with Uyghur data and, then, adapted it to a Kyrgyz model using their private data consisting of 5 h of speech recorded from 40 speakers. Again, the value of these studies is limited as they were not based on an open benchmark.

In 2019, the Common Voice database was published, which contains a small amount of Kyrgyz data. This public data aroused the interest of researchers, and since then, Kyrgyz has become the research object of extremely low-resource language speech recognition [151,152]. For example, Riviere et al. [153] in 2020 verified the performance of transfer learning on these data. They firstly used 360 h of LibriSpeech data to pre-train a contrastive predictive coding (CPC) model and, then, fine-tuned it with 1 h of Kyrgyz data. The performance of the fine-tuned model was 41.2% for the CER. In the same year, Conneau et al. [84] trained a Wav2Vec 2.0 model with 56,000 h of data from 53 languages and, then, fine-tuned it with 1 h of Kyrgyz data, reducing the CER to 6.1%. Furthermore, Baevski et al. [91] proposed a completely unsupervised learning model, which used 1.8 h of unlabeled Kyrgyz data and obtained a CER of 14.9%. Table 6 presents the performance on the Common Voice-Kyrgyz dataset. In this table, CPC and XLSR are "few-shot" systems, where a large amount of extra data were used to pre-train the model, and then, one hour of Kyrgyz data were used to perform the fine-tuning. The pre-trained models are CPC and Wave2Vec2.0, respectively. The last Wav2Vec2.0 system is fully unsupervised, with Wav2Vec2.0 as the feature front-end and a GAN as the decoder.

**Table 6.** SOTA results tested on the Common Voice-Kyrgyz dataset.

| Models | CER (%) |
|---|---|
| Pre-training + fine-tuning (1 h) | |
| CPC [153] | 41.2 |
| Wave2Vec2.0 (XLSR) [84] | 6.1 |
| Completely unsupervised learning (1.8 h) | |
| Wav2Vec2.0 + GAN [91] | 14.9 |

*4.3. Analysis and Discussion*

We briefly summarized the technical developments of the three languages: Uyghur, Kazakh, and Kyrgyz. There are several important observations as follows.

4.3.1. The Importance of Open-Source Data

From the development history of speech recognition technology for the three languages, we can see that, for any language, no matter when the research was started, fast technical progress is impossible without a standard benchmark including both open-source data and reproducible source code.

Taking Uyghur as an example, although the studies on Uyghur ASR started as early as the 1990s, most of the research reported results using private data and without the source codes, making reproduction and technical comparison very hard. The consequence is that research on Uyghur is always in a follow-up state and has little contribution to the mainstream research community.

A standard benchmark is a public platform. On this platform, different technologies can be compared and verified, so it will be easy to verify which technique is effective and which one is a spurious innovation. Famous benchmark databases include TIMIT [154] and WSJ [155] in English and RAS 863 [156] in Chinese. Despite their historical contributions, most of the early datasets are not free. Recently, the open-data movement made a further step: researchers started to publish their data for free and also accompanied by the corresponding source codes for the baselines so that all research groups and individuals can study, communicate, and compare under the same resources and rules. This has significantly promoted technical progress. There are many open-source databases in rich-resource languages, such as LibriSpeech [157], THCHS30 [158], and AIShell-I/II [159,160], to name a few.

The contribution of open-source databases for Uyghur, Kazakh, and Kyrgyz ASR research is even clearer. Especially, we can see that, before the appearance of open-source data, such as THUYG20, KSC, and Common Voice, research on these three languages largely was mostly technique migration from the common languages. However, after the appearance of open-source data, much innovative work emerged. This is the most obvious for the research progress on Kyrgyz. Kyrgyz ASR research started in 2018, much later than the studies on Uyghur. However, due to the Common Voice database, the recognition accuracy has rapidly increased within three years, and Kyrgyz has become one main language for some cutting-edge technologies such as completely unsupervised learning.

The above observations demonstrate that open-source data are perhaps the first and crucial step to promote the technical development of a new low-resource language.

4.3.2. The Diversity of Data Sources

Early databases generally contained reading speech recorded by microphones in an office environment (even high-quality speech recorded in a sound-proof studio [142,143]). Speech data recorded in this way are characterized by clear pronunciation, small noise interference, and a single channel. THUYG-20 and a part of M2ASR-UYGHUR were collected in this way. This type of collection approach is expensive and involves little diversity in the environment and channel. Recently published databases were collected in more flexible ways. For example, most of the data in M2ASR were collected through

mobile apps, which are easy to use and can collect data with more channel and background diversity. KSC and Common Voice adopted a crowdsourcing approach, which allows any person to upload his/her speech and conduct annotation. This greatly speeds up the collection process, reduces the cost, and allows collecting a rich acoustic diversity. Taking Common Voice as an example, from the first version in 2019 to the latest version in 2022, the amount of data has increased 16 times and the number of languages has expanded from 19 to 100. One of the risks of the crowdsourcing scheme is that the annotation may not be accurate enough. However, studies have indicated that, when the data volume is large enough, the influence of annotation noise is acceptable for model training [161,162].

Finally, although the speech data of these three languages are much more diverse than before, they are not truly spontaneous. This is because the speakers will realize that they are being recorded, so they will tend to choose a relatively quiet place and speak clearly. A possible way to collect more spontaneous speech data is to utilize the rich data resources of the Internet. Recently, several public databases are being collected in this way, including GigaSpeech [163], WenetSpeech [164], VoxCeleb [165], CN-Celeb [166], and so on.

### 4.3.3. Language Individuality Not Fully Used

From the technical history of the three languages, it can be seen that the most-significant performance improvement has been due to the evolution of acoustic models developed in common languages, such as the DNN-HMM, Transformer, etc. Even for the techniques developed for low-resource language speech recognition, such as transfer learning and pre-training, they are not specific to Uyghur or Kazakh. At present, we have not found much improvement being attributed to a "special design" for the three languages (using morphemes as LM units could be seen as a special treatment for agglutinative languages, but subword units are also known in English ASR).

This observation should be interpreted from two perspectives. On the positive side, this indicates that it is indeed possible to implement low-resource language speech recognition by improving the general speech recognition technology; at least the most-effective techniques at present can be applied to various languages with no difference. This brings hope to overcome human language barriers: it is not necessary for us to study 7000+ languages one by one, and we can only select some representative languages to develop the general technologies and, then, generalize them to other languages. On the negative side, this means that the language individualities of Uyghur, Kazakh, and Kyrgyz have not been seriously considered in the current research. For instance, we know that people write words in the way that they want to pronounce them, due to a simple letter-to-phoneme mapping. However, pronunciation may vary significantly due to accents or personal habits, leading to uncertainty in word forms. This has caused serious problems in language modeling, but little has been done to resolve it. Other language individualities such as harmony rules, vowel weakening patterns, and phonological constraints have not been well treated in the present research.

### 4.3.4. Language Commonality Not Fully Used

From the technical summary, we can see that the ASR research on Uyghur, Kazakh, and Kyrgyz has almost been independently developed. However, we know that the three languages belong to the same language family and that they share many commonalities. Unfortunately, this commonality is seldom utilized in ASR research. A few exceptions include: Ablimit et al. [132], who developed a general morpheme analysis method based on the agglutinative property of Altaic languages; Sun et al. [150], who transferred a Uyghur model to a Kyrgyz model, assuming that the transfer between languages in the same language family is easy. In spite of these few exceptions, the commonality of the three languages has been ignored by most researchers working on each language.

One possible reason for this situation is the lack of "homologous data". Notice that THUYG-20, KSC, and Common Voice are the most commonly used databases right now for the three languages, respectively. These databases come from different sources and possess

different properties in their quantity, quality, and acoustic features. The discrepancy among the datasets prevents a comparative or collaborative study.

From this point of view, the M2ASR corpus may provide a valuable solution, as the data of the three languages were collected using similar data collection pipelines, so the quality and speaking style are similar. Therefore, we expect the M2ASR data to be suitable for conducting a comparative analysis among the three languages and performing language collaborative learning.

## 5. Conclusions and Discussions

Uyghur, Kazakh, and Kyrgyz all belong to the same language family and branch. Each language has its development path, but they also interact with each other. Especially in China, these three languages share many commonalities in terms of writing, pronunciation, and syntax [2]. From the perspective of speech recognition, they all belong to low-resource languages and encounter similar challenges. The purpose of this paper was to summarize the speech recognition technologies developed for these three languages, with the hope to identify critical problems and effective methods shared among all three languages. Meanwhile, by comparing the technical development pathways of the three languages, we hope to discover important factors for developing low-resource language speech recognition technology.

We first reviewed the technologies that are commonly used to tackle the low-resource language ASR problem and, then, discussed the commonalities and individualities of Uyghur, Kazakh, and Kyrgyz. After that, we summarized the development history and presented the status of speech recognition technologies for the three languages.

Through the discussion, we firstly found that the technical development of these three languages is rather unbalanced. The research progress of Uyghur has been going on for nearly 30 years, while the research on Kyrgyz started only in the last five years. Importantly, regardless of how much sooner or later the research started, rapid progress is always accompanied by the emergence of open-source data.

Secondly, we also observed that the current progress in the three languages is largely attributed to the technology migration from common languages, rather than the "native research" on the target language. On the one hand, this indicates that human languages are similar in pronunciation, and this similarity can be utilized to build a strong ASR system even for languages with very limited resources. On the other hand, this indicates that the current research has paid little attention to languages' individualities.

Thirdly, although the performance of speech recognition on the three languages has greatly improved, we should notice the limitation of the current benchmarks: most of the datasets are comprised of reading speech with low noise. The next technical breakthrough should be accompanied by the emergence of more complex and spontaneous datasets.

Fourthly, there is little research on the uncertainty of the language itself. This uncertainty may be caused by many factors, such as the lack of standardization of the language itself, pronunciation discrepancy in different regions, linguistic changes influenced by word borrowing and general interference from other languages, and so on. Actually, since our focus in the paper was the three languages in China, the observations could be different from those found for the same languages in the middle of Asia. For example, the language description of Kazakh we presented in Appendix B is quite different from recent accounts by other authors [167,168], although both were carefully designed and checked by linguists. Besides, for Uyghur, Kazakh, and Kyrgyz, based on the one-to-one letter-to-phoneme mapping, uncertainty in pronunciation will be reflected in the writing system, leading to noise in the vocabularies and language models. It is worth noting that this uncertainty is a common problem of many low-resource languages, but little research has been conducted in this direction.

In summary, we think that low-resource language speech recognition has three main difficulties: data scarcity, uncertainty in pronunciation and writing, and language individuality. Currently, most studies focus on solving the data scarcity problem, including

more data collection, transfer learning, pre-training, etc. These techniques are obviously of great significance. However, research on the uncertainty and uniqueness of languages is still limited.

For future work, we believe that the following directions are worth exploring: (1) Uyghur, Kazakh, and Kyrgyz should learn from each other's latest technical advances, to construct more powerful ASR systems; (2) research on collaborative modeling methods based on the commonality of the three languages should be conducted; (3) datasets that can better reflect the real-life complexity and verify the true performance of practical systems should be constructed; (4) research on modeling methods for the uncertainty in writing and pronunciation, such as word normalization methods [132] and language model calibration methods (map language models from one region to another region), etc., should be conducted; (5) research on how to utilize language's individualities to further improve system performance should be conducted.

Finally, we noticed that low resources are a general problem in machine learning, not solely in speech recognition. There is a wealth of research on this problem in a broad range of research fields including computer vision, natural language understanding, and robotics. Due to the space limitation, reviewing novel methods in other fields was out of the scope of the paper; nevertheless, we highly recommend researchers in speech recognition look into other fields and, if possible, borrow new ideas and useful tools from them.

**Author Contributions:** Conceptualization, D.W., L.L. and A.H.; methodology, D.W.; software, W.D.; validation, W.D., Y.M. and M.N.; formal analysis, D.W.; investigation, W.D. and Y.M.; resources, W.D. and Y.M.; data curation, W.D. and Y.M.; writing—original draft preparation, D.W.; writing—review and editing, L.L. and W.D.; visualization, L.L. and W.D.; supervision, D.W. and A.H.; project administration, W.D.; funding acquisition, D.W., L.L. and A.H. All authors have read and agreed to the published version of the manuscript.

**Funding:** This research was funded by NSFC grant number 62171250 and 61633013.

**Institutional Review Board Statement:** Not applicable.

**Informed Consent Statement:** Not applicable.

**Data Availability Statement:** Not applicable.

**Conflicts of Interest:** The authors declare no conflict of interest.

## Appendix A. Uyghur Phoneme Table

|            | e          | ɑ          | ø           | i          |
|------------|------------|------------|-------------|------------|
| front/back | front      | back       | front       | front      |
| high/low   | low        | low        | low         | high       |
| Round lips | Lip-opened | Lip-opened | Lip-rounded | Lip-opened |
|            | o          | u           | ε          | y           |
| front/back | back       | back        | front      | front       |
| high/low   | low        | high        | high       | high        |
| Round lips | Lip-rounded | Lip-rounded | Lip-opened | Lip-rounded |

**Figure A1.** Uyghur vowels.

|  |  | Labial | Dental | Palatal | Velar | Uvular | Glottal |
|---|---|---|---|---|---|---|---|
| Stop | voiceless | p | t |  | k | q |  |
|  | voiced | b | d |  | g |  |  |
| Affricates | voiceless |  |  | t͡ʃ |  |  |  |
|  | voiced |  |  | d͡ʒ |  |  |  |
| Fricative | voiceless | f | s | ʃ |  | x | h |
|  | voiced | w | z | ʒ |  | ʁ |  |
| Approximant |  |  | l | j |  |  |  |
| Trill |  |  | r |  |  |  |  |
| Nasal |  |  | m | n |  | ŋ |  |

**Figure A2.** Uyghur consonants.

## Appendix B. Kazakh Phoneme Table

|  | e | æ | ɑ | ø | ɨ |
|---|---|---|---|---|---|
| front/back | front | front | back | front | central |
| high/low | low | low | low | low | high |
| Round lips | Lip-opened | Lip-opened | Lip-opened | Lip-rounded | Lip-opened |
|  | o | u | ə | y |  |
| front/back | back | back | central | front |  |
| high/low | low | high | high | high |  |
| Round lips | Lip-rounded | Lip-rounded | Lip-opened | Lip-rounded |  |

**Figure A3.** Kazakh vowels.

|  |  | Labial | Dental | Palatal | Velar | Uvular | Glottal |
|---|---|---|---|---|---|---|---|
| Stop | voiceless | p | t |  | k | q |  |
|  | voiced | b | d |  | g |  |  |
| Affricates | voiceless |  |  | t͡ʃ |  |  |  |
|  | voiced |  |  | d͡ʒ |  |  |  |
| Fricative | voiceless | f | s | ʃ |  | x | h |
|  | voiced | v | z |  |  | ʁ |  |
| Approximant |  |  | l | j | w |  |  |
| Trill |  |  | r |  |  |  |  |
| Nasal |  |  | m | n |  | ŋ |  |

**Figure A4.** Kazakh consonants.

## Appendix C. Kyrgyz Phoneme Table

|  | e | ɑ | ø | ɨ |
|---|---|---|---|---|
| front/back | front | back | front | front |
| high/low | low | low | low | high |
| Round lips | Lip-opened | Lip-opened | Lip-rounded | Lip-opened |
|  | o | u | ə | y |
| front/back | back | back | central | front |
| high/low | low | high | high | high |
| Round lips | Lip-rounded | Lip-rounded | Lip-opened | Lip-rounded |

**Figure A5.** Kyrgyz Vowels.

| | | Labial | Dental | Palatal | Velar | Uvular |
|---|---|---|---|---|---|---|
| Stop | voiceless | p | t | | k | q |
| | voiced | b | d | | g | |
| Affricates | voiceless | | | t͡ʃ | | |
| | voiced | | | d͡ʒ | | |
| Fricative | voiceless | f | s | ʃ | | x |
| | voiced | w | z | | | ʁ |
| Approximant | | | l | j | | |
| Trill | | | r | | | |
| Nasal | | m | n | | ŋ | |

**Figure A6.** Kyrgyz consonants.

## Appendix D. Differences in Arabic Character Sets of the Three Languages

| ف | ٴۇ | ﺋ | گ | م | ل | ت | ن | پ | ب |
|---|---|---|---|---|---|---|---|---|---|
| ز | ر | د | ۋ | ق | ش | س | چ | ج | ي |

**Figure A7.** Common characters.

| Uyghur | خ | غ | ژ | ﻩ |
|---|---|---|---|---|
| Kazakh and Kirghiz | ح | ع | | |

**Figure A8.** Different characters.

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
