# Peer review of "Automatic Speech Recognition for Uyghur, Kazakh, and Kyrgyz: An Overview"

_applsci, doi:10.3390/app13010326_

Round 1

Reviewer 1 Report

The paper deals with the development and improvement of specific speech technologies (esp. ASR) for under-resourced languages.

Spotted languages are particularly interesting and deep learning techniques which are fostered by the research here described are encouraging (as well as the proposal of a unique M2ASR database).

The paper approaches a relevant issue at this purpose in lines 425-426 or line 484: "the claimed experimental results cannot be reproduced, making the research unverifiable" given that the research reported using private data makes "reproduction and technical comparison very hard".

And that is a general point. However the problem is not only the inaccessible database. The blend of techniques which are used in these studies is quite impressive and the way they are combined in the final procedure is not always explicitely shown. It is then commonly assumed that the reader must rely on the quality judgement discussed by the same authors of the experiment.

Another issue which is addressed (without a specific solution) is related to language variation that raises uncertainty (lack of standardization of the language, pronunciation variation, linguistic changes influenced by word-borrowing and general interference by other languages). That may be the reason why language description in the Appendix (e.g. Kazakh) differ so much from recent accounts by other authors (see McCollum and Chen Journal of the International Phonetic Association , Volume 51, Issue 2, August 2021, pp. 276 – 298, and Özçelik, Öner, “Kazakh phonology”, In L. Johanson, E.A. Csato Johanson, L. Karoly & A. Menz (eds.), Encyclopedia of Turkic Languages and Linguistics, Leiden: Brill, 6 pages, https://oozcelik.pages.iu.edu/papers/Kazakh%20phonology.pdf).

-------------------------------------------------------------------

As a general remark, if one writes "Kyrgyzstan" he/she is expected calling "Kyrgyz" the language (elsewhere one may prefer Kirghiz and Kirghizistan, as French people do, on more classical bases - "Kirghiz" is everywhere in the paper: I then suggest to replace Kyrgyzstan).

9 AItaic > Altaic

31-33 consider reformulating the list in a different way from the abstract. Especially "individuality of each language" can be better explained

38 AItaic > Altaic

123 vocal track > vocal tract

121-126 In my personal research activity I am not concerned by Data augmentation techniques and I am a bit confused here. I wander how "vocal tract length" may be perturbated in a recorded speech sample. A few words of explanation could help the reader.

194 unlabeled > unlabelled (see 201)

198 pseudo labels > pseudo-labels (idem 214-215 extra text > extra-text, extra training > extra-training)

208 40 Hindi languages. Needs to be reformulated ("40 languages spoken in India" or "40 dialects of the Hindi language"?)

226 GAN model you may add here a short presentation of the model.

234 JHU: expand the abbreviation

249-250 a broad estimation of number of speakers using the three languages is welcome here (this data may be introduced in lines 40-42)

254, 265, 290, 291, 294, 299, 558 AItaic > Altaic

256, 346, 352 Kirgiz > Kirghiz (but see above, also see 108 and 110: check how Abulhasm and his title are transliterated -Abu 'l-hasn?- The same applies for Yishak's title)

280-282 the formulation is too much vague (F0 patterns are similar... How F1 may be the same? You probably meant that some formants estimated on a large speech sample tend to stay around the same values: that depends on how many male and female speakers are taken into account...)

302 I would not say "contains" but "is based on"

304 "was substituted for" > "was substituted WITH"

315 check italics here

322 spirantization of semivowel y > spirantization of the semivowel spelled as a letter transliterated with y ([j]?)

335-336 how many female speakers?

404 BLSTM? Even though most shortenings are well known in the field, expanding the abbreviaton is welcome here (as is done in 134 for IPA and 212 MMDA; also check 176 SGMM, 185 LF-MMI, 445 LSTM, LM in Table 4)

423 1000 hours of speech > about (?) 1000 hours of speech)

587 imbalanced > unbalanced

814 "Proceedings of the Proceedings of the" > "Proceedings of the"

836 asr > ASR

841 karlsruhe > Karlsruhe

...

I don’t know if the following basic representational issue may bias all the experiments in this field, but I think that a relible description should be at the bases of a thorough assessment.

Table 3 in Appendix B: vowels slightly differ from the ones described by McCollum & Chen (2021). Even assuming a different notation for the same vowel sounds, they include two other sounds (broadly a /U/, contrasting with /u(w)/, and /ij/, contrasting with /I/). A discussion about that issue should include an assessment of Özçelik’s paper.

Table 4 in Appendix B:

Nasel > Nasal

Kazakh consonants described in McCollum & Chen (2021) are quite different.

Besides Palatal j, only postalveolar S Z are listed: no palatal affricates are confirmed (and only an affricate is accounted for by Özçelik). Therefore, at least palatals should be re-analyzed and the table updated or better discussed.

Furthermore McCollum & Chen (2021: 277) describe [h] as an occasional variant and not a phoneme (the same is in Özçelik).

Table 8 in Appendix: difference characters > different characters

Reviewer 2 Report

1. Pseudocode / Flow Chart and algorithm steps need to be inserted.
2. The experimental results need to be explained in more details and what is the accuracy of the proposed method.

3. precision recall f1 measure need to be calculated.
4. Limitation and discussion Sections need to be inserted.

Round 2

Reviewer 1 Report

I think you made the necessary integrations and correctly revised the paper. Congratulations. It looks nice for me now.

Only line 287 now has "languges" instead of "languages"

Author Response

Dear Reviewer:

Thank you for reminding us that we have corrected this error.

Thanks again

ours sincerely,
Dong Wang
on behalf of the co-authors

Reviewer 2 Report

The review comments are not fixed.

Author Response

First of all, thanks for the comments suggested by reviewer. 

We have incorporated the reviewer's comments 1 and 3, and all the major revisions have been highlighted by red text in the manuscript.
We also have carefully considered the reviewer's concerns 2 and 4, and provided our explanation in the response below.

Again, really appreciate the valuable comments – which are very instructive for us to polish the paper.

Reviewer:
1. Pseudocode / Flow Chart and algorithm steps need to be inserted.

Thanks for the suggestion.

Actually, the contribution of this paper is to overview the speech recognition techniques devel-oped for Uyghur, Kazakh, and Kirghiz, with the purpose (1) to highlight the important tech-niques that have been broadly verified so that researchers working on different languages canlearn from each other; (2) to discover the key factors in promoting speech recognition research for low-resource languages, thereby providing some suggestions on the ‘important work’ that researchers may focus on. We therefore focus more on the technical development roadmap of the three languages rather than the technical details of a particular algorithm.
Again thanks for the suggestion and we will consider it in future (e.g., reproduce all the stuff based on M2ASR and report a full-fledged empirical verification).
2. The experimental results need to be explained in more details and what is the accuracy of the proposed method.

Thanks for the suggests.

We intentionally refrained us from providing too much details, to avoid jeopardizing the ‘global view’ of the paper. However we admit that some sentences to explain what techniques each system uses will ease the reading. We add the explanation sentences in the revision.
3. precision recall f1 measure need to be calculated.

In fact, Character Error Rate (CER) and Word Error Rate (WER) are the two most common metrics of the performance of automatic speech recognition, and precision, recall, and f1 are not often used (more frequently used for keyword spotting?).
4. Limitation and discussion Sections need to be inserted.

Thanks for your suggestion. We changed Section ‘Conclusions’ to ‘Conclusions and Discus-sions’. We also added the following sentences to highlight the limitation of the overview.
“ Finally, we notice that low-resource is a general problem in machine learning, not solely for speech recognition. There are a wealth of research on this problem in a broad research fields including computer vision, natural language understanding, robotics. Due to the limited pages, reviewing novel methods in other fields is out of the scope of this paper; nevertheless, we highly recommend researchers in speech recognition pay an eye on other fields, and if possible borrow new ideas and useful tools there. ”

Round 3

Reviewer 2 Report

Accept after adding a flowchart and steps of the proposed study.

Author Response

Dear Reviewer:

Thank you for your advice,

 We added a table in Section 2.2 to illustrate the representative methods of low-resource ASR.

Thanks again

ours sincerely,
Dong Wang
on behalf of the co-authors
